# Fungal mycelia and bacterial thiamine establish a mutualistic growth mechanism

Gayan Abeysinghe*, Momoka Kuchira*, Gamon Kudo, Shunsuke Masuo, Akihiro Ninomiya, Kohei Takahashi, Andrew S Utada, Daisuke Hagiwara, Nobuhiko Nomura, Naoki Takaya, Nozomu Obana, Norio Takeshita

Exclusivity in physical spaces and nutrients is a prerequisite for survival of organisms, but a few species have been able to develop mutually beneficial strategies that allow them to co-habit. Here, we discovered a mutualistic mechanism between filamentous fungus, *Aspergillus nidulans*, and bacterium, *Bacillus subtilis*. The bacterial cells co-cultured with the fungus traveled along mycelia using their flagella and dispersed farther with the expansion of fungal colony, indicating that the fungal mycelia supply space for bacteria to migrate, disperse, and proliferate. Transcriptomic, genetic, molecular mass, and imaging analyses demonstrated that the bacteria reached the mycelial edge and supplied thiamine to the growing hyphae, which led to a promotion of hyphal growth. The thiamine transfer from bacteria to the thiamine non-auxotrophic fungus was directly demonstrated by stable isotope labeling. The simultaneous spatial and metabolic interactions demonstrated in this study reveal a mutualism that facilitates the communicating fungal and bacterial species to obtain an environmental niche and nutrient, respectively.

## Introduction

Microbes ubiquitously live in nearly every ecological niche. Different species coexist in certain habitats and interact with each other. Microbes often constitute communities and share available metabolites (Romine et al, 2017). Natural auxotrophic strains grow in the presence of external nutrients which are provided by members of the local microbiota (Zengler & Zaramela, 2018). Because such nutrients limit microbial growth, acquiring them within communities is essential for auxotrophs to use an ecological niche.

Bacteria and fungi comprise a large fraction of the biomass in soil (Osono et al, 2003; Fierer, 2017). Because they interact with each other to carry out their characteristic functions in the ecosystem, a better knowledge of bacterial–fungal interactions is important for understanding the microbial ecosystem, which is closely related to agriculture, medicine, and the environment (Nazir et al, 2010). Inter-kingdom interactions are driven by diverse factors such as anti-biotics, signaling molecules, cooperative metabolism, and physical interactions (Frey-Klett et al, 2011). In certain scenarios, bacteria physically attach to fungal tube-shaped hyphal cells, thus enabling changes in their metabolism either antagonistically or beneficially (Benoit et al, 2015). It has been shown that fungal hyphae transfer nutrients and water to activate bacteria (Worrich et al, 2017), whereas bacteria are able to induce the expression of transcriptionally inactive genes for synthesizing fungal secondary metabolites (Nutzmann et al, 2011).

Filamentous fungi grow by extension of hyphae at their tips, thereby forming multi-cellular networks with branching cells at subapical regions (Takeshita, 2016; Riquelme et al, 2018). Mycelial network spreads on solid surfaces that allow the fungus to reach spatial niches in the ecosystem. In contrast, bacteria are unicellular organisms, some of which are motile, enabling them to explore the environment in search of better spatial and nutrient conditions (Kearns, 2010). Although motility is efficient in liquid (Harshey, 2003), bacteria can disperse farther in water-unsaturated conditions by traveling along fungal hyphal "highways" (Kohlmeier et al, 2005; Pion et al, 2013). This interaction is considered as commensal because the fungi do not benefit from providing a "highway" for bacteria.

Here, we describe a mutualistic growth mechanism between models of filamentous fungus, *Aspergillus nidulans* (Takeshita et al, 2017; Riquelme et al, 2018; Zhou et al, 2018) and gram-positive bacterium *Bacillus subtilis* (Cazorla et al, 2007), where both organisms benefit from the fungal highways and the sharing of one vitamin.

## Results

### Bacterial movement along fungal hyphae

We tested several combinations of fungal–bacterial co-culture among our laboratory strains, then selected the combination of

Microbiology Research Center for Sustainability (MiCS), Faculty of Life and Environmental Sciences, University of Tsukuba, Tsukuba, Japan

Correspondence: obana.nozomu.gb@u.tsukuba.ac.jp; takeshita.norio.gf@u.tsukuba.ac.jp
Nozomu Obana's present address is MiCS and Transborder Medical Research Center, Faculty of Medicine, University of Tsukuba, Tsukuba, Japan
*Gayan Abeysinghe and Momoka Kuchira contributed equally to this work

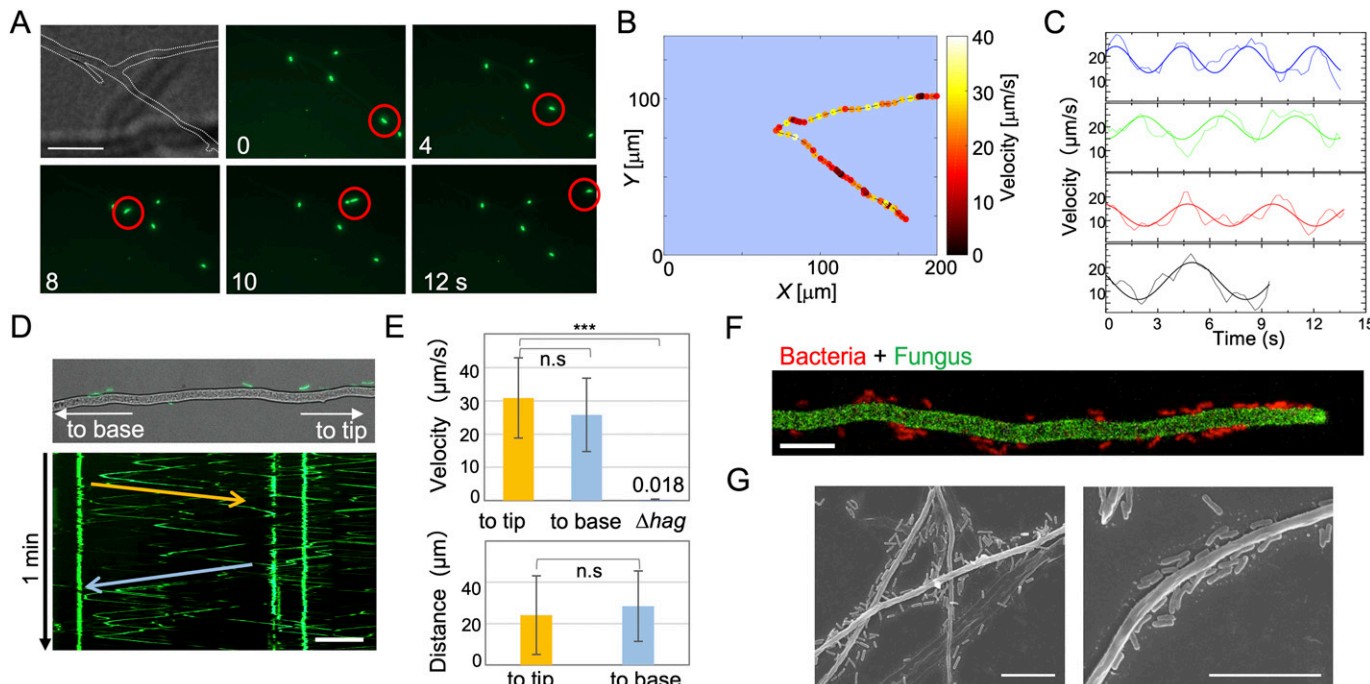

**Figure 1.  Bacterial movement along fungal hyphae.**
**(A)** Time-lapse images of *B. subtilis* (expressing green fluorescence ZsGreen) movement along *A. nidulans* hyphae (dotted line) for 30 s on agar media from Video 1. Scale bar: 50 *μm*. **(A, B)** Heat map of *B. subtilis* instantaneous velocity analyzed by tracking the position of the cell moving along the hypha in (A). **(B, C)** Weak oscillations in the instantaneous velocity over time are shown in different colors from each bacterial cell in (B) and Fig S2. **(D)** Kymograph of *B. subtilis* movement along the hyphae (top) to the tip (yellow arrow) and base (blue arrow) from Video 2. Total 1 min. Scale bar: 50 *μm*. **(E)** Velocity and distance of *B. subtilis* (wild-type or Δ*hag*) movement along hyphae to the tip (yellow) or base (blue). Error bar: SD, n = 26 (to tip), 28 (to base), 5 (Δ*hag*), ***P ≤ 0.001. **(F)** *B. subtilis* cells (red) reach the tip of *A. nidulans* hyphae (green) from Video 3. Scale bar: 10 *μm*. **(G)** SEM images of co-culture of *B. subtilis* and *A. nidulans.* Scale bar: 10 *μm*.

*A. nidulans* and *B. subtilis*, that are relatively common in soil, for further analysis. *B. subtilis* grew in co-culture with *A. nidulans* at the comparable rate as in liquid monoculture (Fig S1A and B). Live imaging analysis showed that *B. subtilis* cells moved along the co-cultured *A. nidulans* hyphae on the agar medium (Fig 1A and Video 1). Some bacteria remained attached to the hyphae, whereas others moved along the hyphae, often reversing course abruptly and beginning to move in the opposite direction. A heat map of the instantaneous velocity was constructed by tracking the positions of each moving cell; the results indicated weak oscillations in the instantaneous velocity over time (Figs 1B and C and S2). Kymographs indicated that the bacterial cells moved at an average velocity of ~30 *μm* s$^{-1}$ in both directions (Fig 1D and E and Video 2). Rapid movements were not observed in bacterial monoculture on solid medium (Fig S3A) and were comparable with *B. subtilis* movement in fungus-free liquid medium (Matsuura et al, 1977). The numbers and density of motile and non-motile bacteria were not uniform on the mycelium. Some bacterial cells reached the hyphal tips and then reversed their direction after remaining at the tip for some time (Fig 1F and Video 3). Other hyphae were surrounded by moving bacterial aggregates (Fig S3B and Video 4). The co-cultures were observed by a scanning electron microscope (Fig 1G).

B. subtilis strain 168, which is defective in producing the biosurfactant surfactin necessary for swarming on solid agar plates (Kearns & Losick, 2003), still moved along the hyphae. In contrast, the flagellar-deficient mutant (Δ*hag*) traveled toward the hyphal

tips at considerably lower rates (0.018 *μm* s$^{-1}$, 0.05% of control strain) (Figs 1E and S3C and Video 5), indicating that *B. subtilis* move along the hyphae using flagella.

### Bacterial dispersal on growing fungal hyphae

*B. subtilis* generated smaller colonies on the agar medium than *A. nidulans* (Fig 2A). The size of the co-cultured *A. nidulans* colony was ~30% larger than that of fungal monoculture. Fluorescence-tagged *B. subtilis* were observed on the colony periphery of the co-culture. This dispersal depended on bacterial movement toward the hyphal tips (Fig 2B and Video 6). The rate of bacterial colony expansion was sevenfold faster (154 ± 33 *μm* h$^{-1}$) than that of *B. subtilis* monoculture (21 ± 3 *μm* h$^{-1}$) (Fig 2B and C and Video 7). The flagella were indispensable for bacterial dispersal along the growing hyphae and for the cells to reach the hyphal tips (Fig 2A–C and Video 8). The colony expansion rate of the Δ*hag* strain was comparable with that in the wild-type monoculture (Fig S4). Because the bacterial movement along the hyphae was much faster (~30 *μm* s$^{-1}$) than the extension rate of growing hyphae, bacterial cells reached the ends of the hyphal tips (Fig 2D and Video 9). The mycelium network appears to supply a space for bacteria to migrate, disperse, and proliferate (Fig 2E and Video 10). Indeed, the bacterial proliferation measured by the amount of bacterial DNA was higher in the co-culture more than the monoculture on the agar plates (Fig S1C).

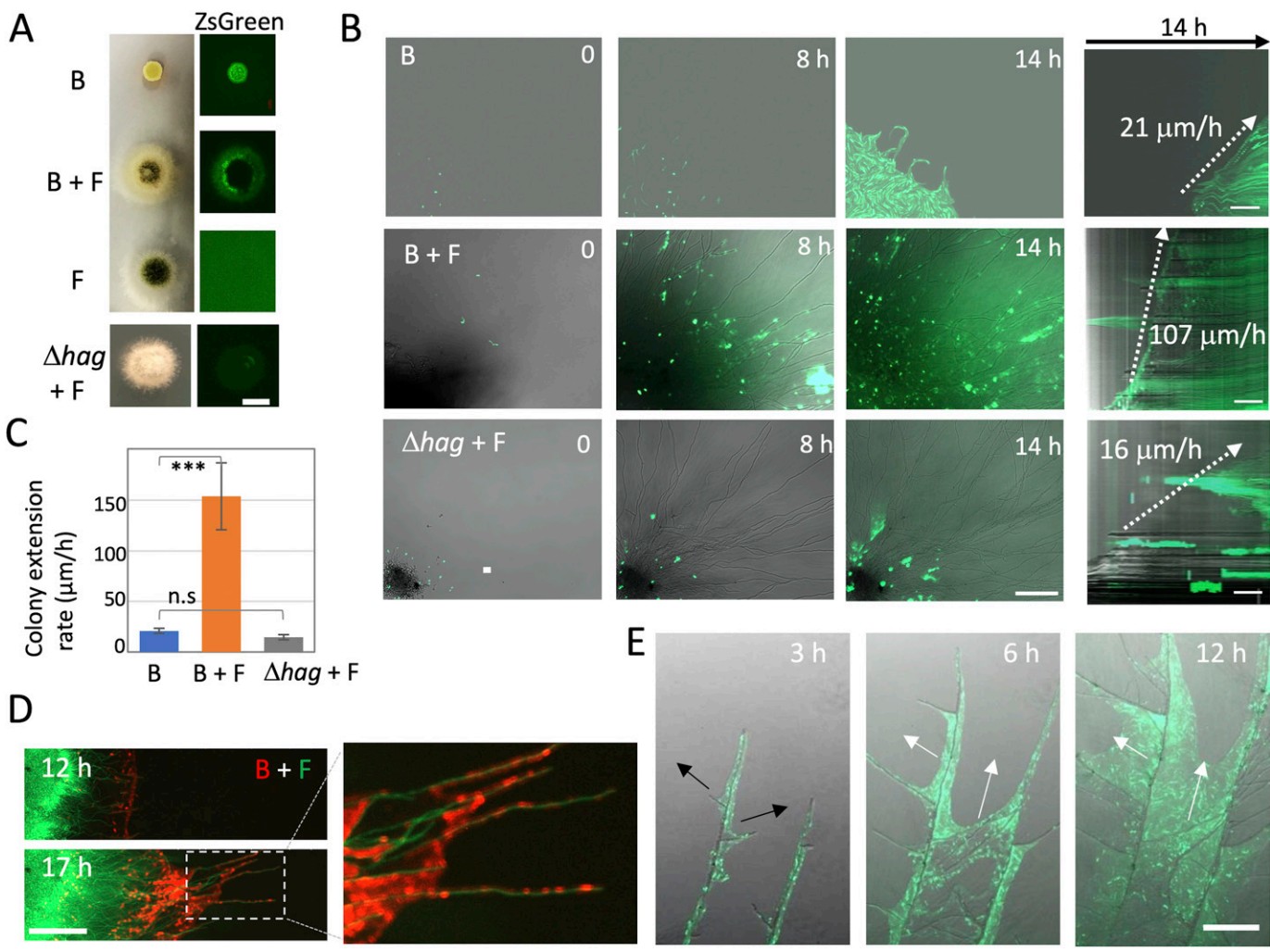

**Figure 2. Bacterial dispersal on growing fungal hyphae.**
**(A)** Colonies of *B. subtilis* and *A. nidulans* monoculture and co-culture, co-culture of *A. nidulans* and *B. subtilis* (Δ*hag*) (bottom). ZsGreen-labeled *B. subtilis* (right). Aerial growing hyphae at the middle of the colony disturb to detect the fluorescent signals in the co-culture. B, *B. subtilis*; F, *A. nidulans*. Scale bar: 5 mm. **(B)** Time-lapse images at 0, 8, and 14 h of *B. subtilis* monoculture, *B. subtilis* (WT or Δ*hag*) with *A. nidulans* from Videos 6–Videos 8. Scale bar: 100 *μm*. Kymographs of *B. subtilis* (WT or Δ*hag*) dispersion with/without growing hyphae (right). The dotted arrows indicate the velocity of dispersal of *B. subtilis*. Scale bar: 50 *μm*. **(C)** Colony expansion rates calculated from kymographs. Error bar: SD, n = 5, ***P ≤ 0.001. **(D)** Time-lapse *B. subtilis* (red) dispersion on growing *A. nidulans* hyphae (green) from Video 9. Scale bar: 200 *μm*. **(E)** Time-lapse *B. subtilis* dispersion (green, white arrows) on *A. nidulans* branching hyphae (DIC, black arrows) after 17 h co-culture from Video 10. Scale bar: 200 *μm*.

## Metabolic interaction through thiamine

We analyzed the effect of co-culture of *B. subtilis* and *A. nidulans* on extracellular hydrophobic metabolites (Fig S5) and transcriptomes. RNA-sequencing analysis indicated that expression of most *B. subtilis* and *A. nidulans* genes was not affected by the co-culture. The expression of 18 genes in *B. subtilis*, including the thiamine biosynthesis operon, was induced by twofold in the co-culture with *A. nidulans* (Figs 3A and S1D and Table S1). In contrast, thiamine biosynthesis-related genes in *A. nidulans* were down-regulated in co-culture (Fig 3A and Table S2). The up-regulated genes in *A. nidulans* include asexual spore formation, nitrate inducible genes, non-ribosomal peptide synthases, and polyketide synthases (Table S3). The induction in *B. subtilis* and repression in *A. nidulans* of thiamine biosynthesis–related genes implied that the bacterium and the fungus metabolically interact via thiamine. We co-cultured

the *A. nidulans* strain defective in thiamine biosynthesis (Δ*thiA*; putative thiazole synthase) (Shimizu et al, 2016) with *B. subtilis*. The Δ*thiA* fungal colony showed a severe growth defect on the plate without thiamine (Fig 3B), which was recovered by adding thiamine. The growth defect of Δ*thiA* was recovered by co-culture with wild-type *B. subtilis*, but not by co-culture with the *B. subtilis* thiamine synthesis mutant (Δ*thi*; operon deletion) (Fig 3B), indicating that *B. subtilis* synthesizes and supplies thiamine to *A. nidulans*.

The wild-type *B. subtilis* strain spread to the periphery of the co-cultured fungal Δ*thiA* colony on the plate without thiamine (Fig 3C). The *B. subtilis* Δ*thi* cells also dispersed to the periphery of the fungal Δ*thiA* colony even though the fungal Δ*thiA* colony showed a severe growth defect. Because flagella are required for bacterial dispersal on the hyphae, the Δ*hag* cells grew at the center of the fungal colony but did not reach the periphery of the fungal Δ*thiA* colony (Fig 3C). Notably, the fungal growth defect was hardly

A

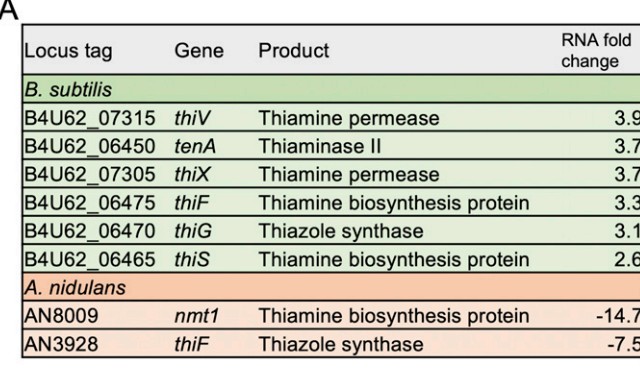

| Locus tag | Gene | Product | RNA fold change |
|---|---|---|---|
| *B. subtilis* | | | |
| B4U62_07315 | *thiV* | Thiamine permease | 3.9 |
| B4U62_06450 | *tenA* | Thiaminase II | 3.7 |
| B4U62_07305 | *thiX* | Thiamine permease | 3.7 |
| B4U62_06475 | *thiF* | Thiamine biosynthesis protein | 3.3 |
| B4U62_06470 | *thiG* | Thiazole synthase | 3.1 |
| B4U62_06465 | *thiS* | Thiamine biosynthesis protein | 2.6 |
| *A. nidulans* | | | |
| AN8009 | *nmt1* | Thiamine biosynthesis protein | -14.7 |
| AN3928 | *thiF* | Thiazole synthase | -7.5 |

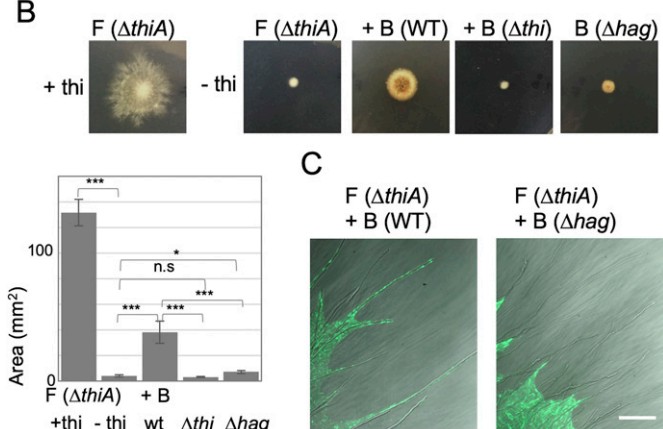

**Figure 3. Metabolic interaction through thiamine.**
**(A)** Summary of RNA-seq analysis related to thiamine synthesis in *B. subtilis* (green) and *A. nidulans* (orange). **(B)** Fungal colonies of *A. nidulans* (ΔthiA) monoculture or co-cultivated with *B. subtilis* (WT, Δthi, or Δhag) on minimal medium with/without thiamine grown for 2 d at 30°C. The area of colonies is measured by ImageJ software. Error bar: SD, n = 3, ***P ≤ 0.001, *P ≤ 0.05. **(C)** Dispersal of *B. subtilis* (WT or Δhag with ZsGreen) on colonies of *A. nidulans* (ΔthiA) without thiamine grown for 2 d at 30°C. Scale bar: 100 μm.

recovered by co-culture with the non-motile Δ*hag* strain (Fig 3B). This was confirmed by other three non-motile mutants (Fig S6A and B). These results indicated that simultaneous bacterial dispersion to the periphery of the fungal colony and supply of thiamine were required for the normal growth of the *A. nidulans* ΔthiA strain.

We measured the amount of thiamine in the supernatant or fungal cell extracts of the co-culture and monoculture by LC-MS-(MRM) multiple reaction monitoring analysis (see the Materials and Methods section). The supernatant of *B. subtilis* monoculture contained thiamine 140 ± 2 ng/ml, whereas thiamine was not detected in that of *B. subtilis* Δthi (Figs 4A and S6C), indicating that *B. subtilis* cells synthesized and secreted thiamine in the medium. The amount of thiamine in the supernatant of co-culture with wild-type *A. nidulans* decreased to 94 ± 5 ng/ml. In contrast, the amount of thiamine in the fungal cell extracts in co-culture with wild-type *B. subtilis* is 134 ± 5 ng/g (wet weight), which was higher than that of fungal monoculture, 95 ± 5 ng/g (wet weight). These support a thiamine transfer from *B. subtilis* to *A. nidulans*.

The *B. subtilis* wild-type or Δthi cells were labeled by culturing in medium containing stable isotope $^{13}$C-glucose, whereas the *A. nidulans* was cultured in medium containing normal glucose. After the 2-d monoculture, the cells were washed and co-cultured. The

LC–MS analysis detected $^{13}$C thiamine in the washed fungal cell extracts in the co-culture with the wild-type *B. subtilis* (Fig 4B), but not in the co-culture with *B. subtilis* Δthi. These results directly demonstrate that *A. nidulans* cells take the thiamine up from *B. subtilis*. Indeed, the fungal biomass in the co-culture was 40% higher than that in the fungal monoculture (Fig 4C), consisting with Fig 2A. In contrast, the amount of thiamine in fungal cell extracts and the fungal biomass did not increase in the co-culture with *B. subtilis* Δthi. These indicate that the supply of thiamine from *B. subtilis* promotes the fungal growth.

We constructed a *B. subtilis* thiamine reporter strain, expressing ZsGreen, under the constitutive promoter, and mScarlet-1, under the control of thiamin pyrophosphate (TPP)-riboswitch, whose expression is activated in the thiamine-depleted condition (Fig 4D) (Mironov et al, 2002). The mScarlet-1 was not expressed in the bacterial colony grown with thiamine but induced without thiamine (Figs 4D and S7A). In co-culture with *A. nidulans* as well, the mScarlet-1 was not expressed with thiamine but induced without thiamine. Notably, the induction was significantly higher at the edge of colony than in the middle (Figs 4E and S7A and B). These indicate that *B. subtilis* cells produce more thiamine at the colony edge because *A. nidulans* takes thiamine up at the growing hyphal tips.

Taken together with our results, bacterial benefit is the bacterial cells moving faster along hyphae and the hyphae delivering the bacteria farther, whereas the fungal benefit is delivery of thiamine to hyphal tips by bacterial cells and resultant promotion of fungal growth.

The LC-MS-MRM analysis indicated the amount of thiamine in the fungal cell extracts in co-culture with *B. subtilis* Δthi was lower than that of fungal monoculture (Fig 4A). CFU of *B. subtilis* in the co-culture of *B. subtilis* Δthi and *A. nidulans* wild-type was higher than that in monoculture of *B. subtilis* Δthi, whereas that in co-culture *B. subtilis* Δthi and *A. nidulans* ΔthiA was comparable with monoculture of *B. subtilis* Δthi (Fig S6D). These results indicate bidirectional thiamine transfer between *B. subtilis* and *A. nidulans*.

### Ecological relevance of mutualistic interaction

To evaluate the ability of fungi to disperse bacteria in nature, we designed a soil-sandwich experiment as follows. A square section of agar with co-cultured *A. nidulans* and *B. subtilis* (right) and another new section of agar (left) were placed a few millimeters apart; the separation between the two sections was filled with soil particles (Fig 5A). The fungal hypha protruding from the right agar continued to extend into the soil particles and eventually reached the left agar slab (Figs 5A and S8A and Video 11). Migration of *B. subtilis* cells (green) followed the mycelium extension, through the soil particles, and to the left agar slab. In the absence of the fungus or soil particles, no bacteria migrated beyond the gap between agar sections (Fig S8B and C and Videos 12 and 13). These indicate that hyphal growth toward favorable nutrient conditions on dry solid substrates enables bacteria to move along the hyphae and explore previously inaccessible spatial niches in nature.

To confirm the ecological relevance, we screened a bacterial–fungal complex, where bacteria moved along the fungal hypha, from natural soil, and co-isolated *Trichoderma* sp. (*harzianum* and *neotropicale*; 100% identity of 255 bp ITS) and *Pantoea* sp. (98.4% similarity to the full length of 16S rRNA gene in *Pantoea rodasii*). The fungus co-cultured with bacteria grew better than the fungus

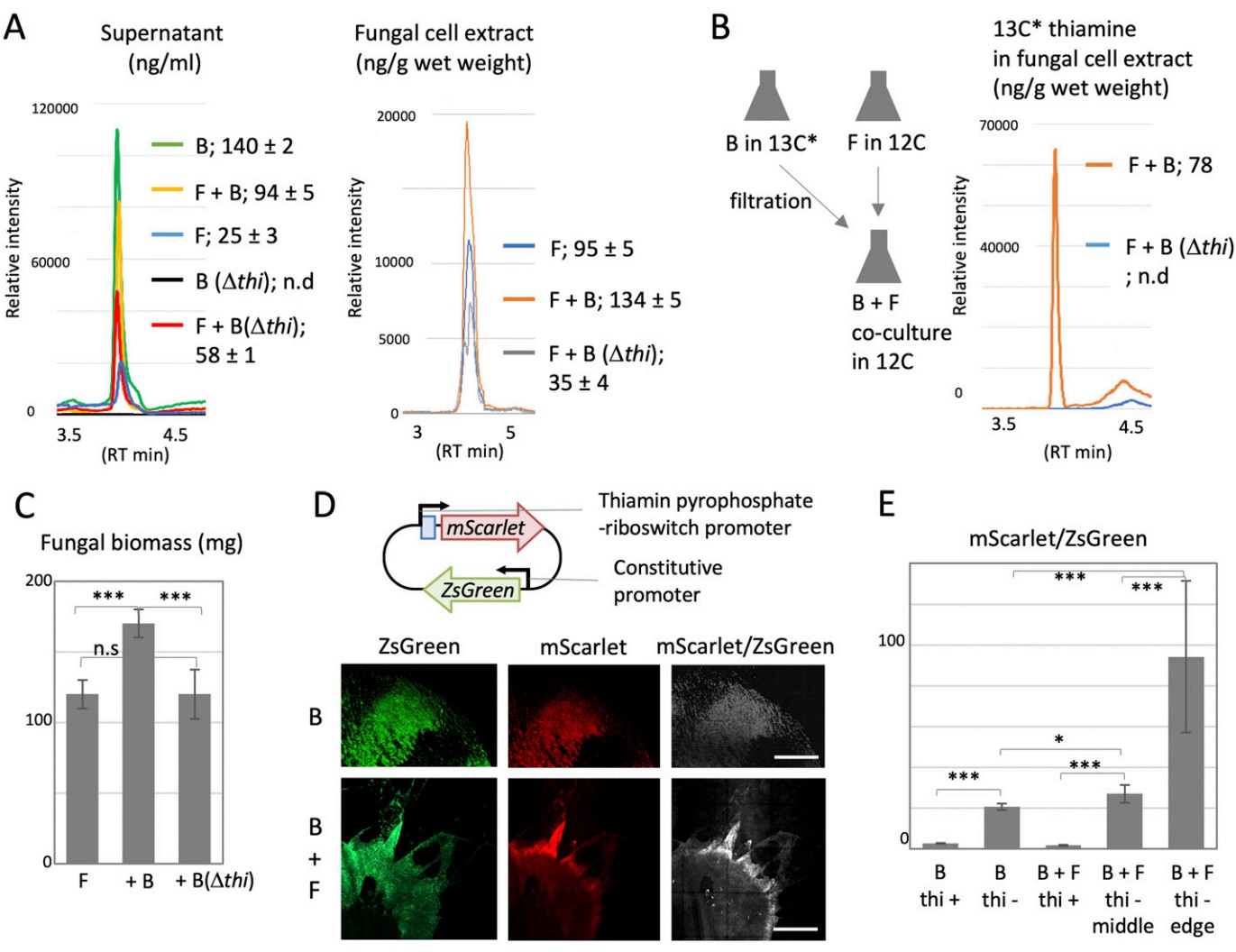

**Figure 4. Thiamine transfer analyzed by molecular mass and reporter strain.**
**(A)** The amount of thiamine in supernatant and fungal cell extracts in the co-culture and monoculture of wild-type *A. nidulans* and *B. subtilis* (WT or Δ*thi*) by LC-MS-MRM analysis. B, *B. subtilis*; F, *A. nidulans*. The mean values of peak and SD are shown. n = 3. **(B)** The amount of $^{13}$C* thiamine by LC-MS-MRM analysis in the fungal cell extracts in the co-culture of *A. nidulans* pre-grown in $^{12}$C and *B. subtilis* (WT or Δ*thi*) pre-grown in $^{13}$C*. **(C)** The fungal biomass in wild-type *A. nidulans* monoculture and co-culture with *B. subtilis* (WT or Δ*thi*). Error bar: SD, n = 3, ***P ≤ 0.001. **(D)** Construct of *B. subtilis* thiamine reporter strain. Colonies of the *B. subtilis* reporter strain monoculture and co-culture with *A. nidulans* on the minimal medium without thiamine grown for 2 d at 30°C. The images are constructed by 10 × 10 tiling of 500 × 500 μm confocal image. Scale bar: 500 μm. **(E)** Ratio of signal intensity, mScarlet-1/ZsGreen, in 500 × 500 μm confocal image normalized by ZsGreen intensity. Error bar: SD, n = 3, ***P ≤ 0.001. *P ≤ 0.05.

monoculture (Fig 5B). In the co-culture, the bacteria cells moved along the hyphae and reached the tips of hyphae (Video 14). The fungal growth was promoted with the addition of thiamine to the whole medium, but not with spot inoculation of bacterial cell lysates, which mimic a non-motile mutant. This example supports the ecological relevance of similar mechanism observed in the co-cultured *A. nidulans* and *B. subtilis*.

# Discussion

Here, we displayed bacterial motility along fungal hyphae and bacterial dispersal on mycelial extension. The phenomena were observed on agar media and in the soil, which are water-unsaturated

conditions. It is likely that water surrounds and covers the hyphae because of surface tension, thus providing space around the hyphae for bacterial cells to swim by their flagella. Mycelium networks extend in natural environment, especially throughout the soil, which could function as roads and bases for bacteria to migrate and proliferate. This is consistent with a recent study showing that fungal networks shape the dispersal of bacteria in the cheese rind microbiota (Zhang et al, 2018).

Previous works have demonstrated bacterial transportation via fungal highway (Kohlmeier et al, 2005; Ingham et al, 2011; Pion et al, 2013), and several metabolic interactions have been analyzed between fungi and bacteria (Deveau et al, 2010; Frey-Klett et al, 2011; Benoit et al, 2015; Worrich et al, 2017). However, these fungal–bacterial interactions are commensal. Our transcriptomic, genetic, molecular mass, and imaging analyses demonstrates that the bacterial cells travel faster along mycelia depending on their

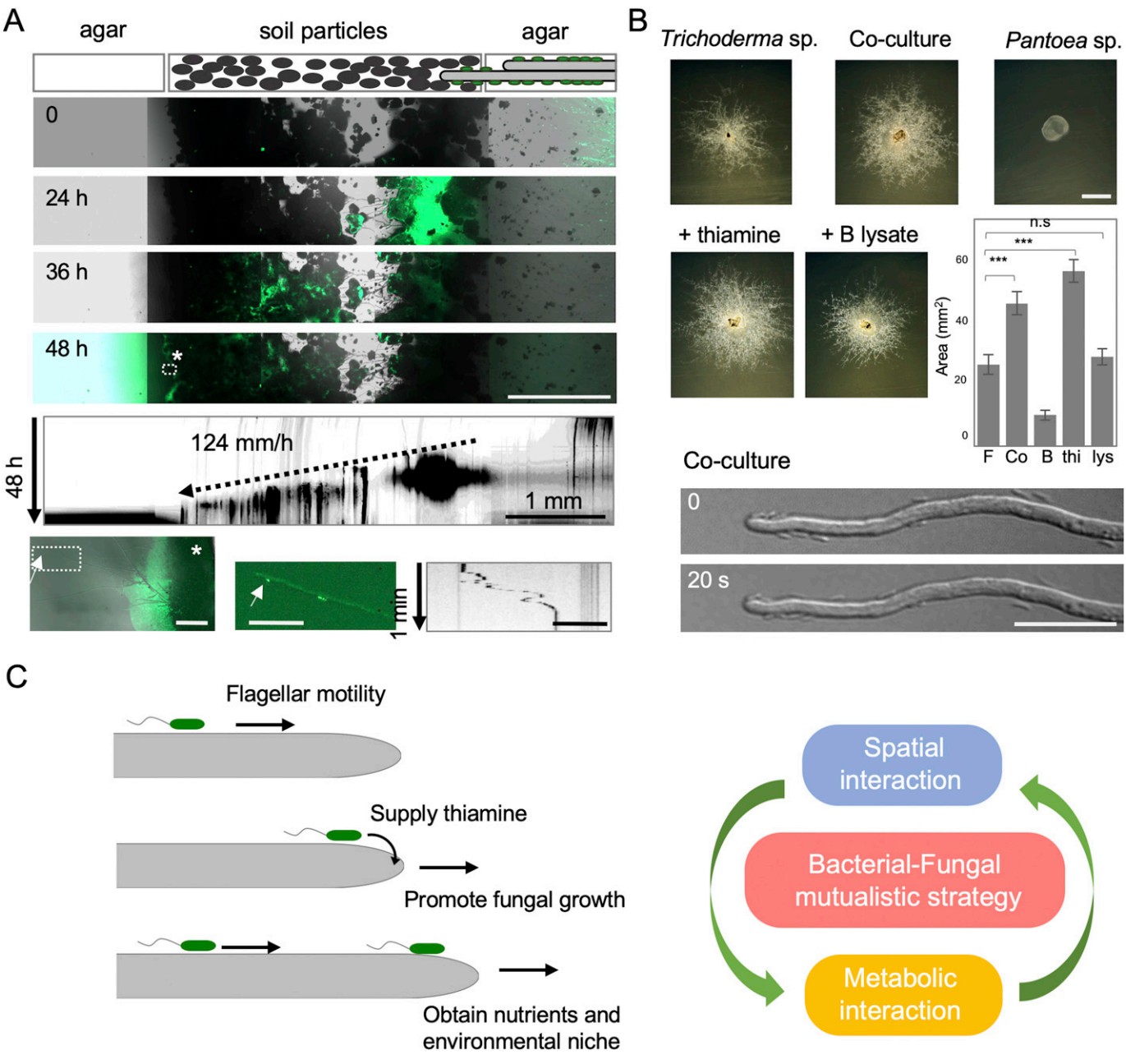

**Figure 5. Mutualistic growth strategy by spatial and metabolic interactions.**
**(A)** Time-lapse bacterial dispersal (green) on growing hyphae in soil particles sandwiched between two agar pieces at 0, 24, 36, and 48 h. Scale bar: 1 mm. Kymograph of bacterial migration from Video 11 (vertical arrow: 48 h, scale bar: 1 mm). Asterisk indicates an expanded image of bacterial colony (green) and mycelium at the left agar after 48 h (left bottom). Scale bar: 100 $\mu$m. Arrows indicate bacterial movement along hyphae at the left agar after 48 h (middle bottom). Scale bar: 50 $\mu$m. Kymograph of the bacterial movement (right bottom). Vertical arrow: 1 min, Scale bar: 50 $\mu$m. **(B)** Colonies *Trichoderma* sp. and *Pantoea* sp. monoculture and co-culture. The bacterial cell lysate is prepared by sonication. Scale bar: 2 mm. The area of colonies is measured by ImageJ software. Error bar: SD, n = 3, ***$P \leq 0.001$. Expanded image of the bacterial cells move along the hyphae and reach the tip. Scale bar: 20 $\mu$m. **(C)** Mutualistic growth strategy that the bacterial cells move faster along fungal highway and disperse farther on fungal growth, whereas bacterial cells supply thiamine to hyphal tips and promote the fungal growth.

flagella and disperse farther with expansion of fungal colony, and at same time, that the bacteria reach the mycelial edge and supply thiamine to the growing hyphae, resulting in a promotion of hyphal growth (Fig 5C). We propose a novel mutualistic growth mechanism in bacterial–fungal interactions that the bacterial cells move along the fungal highway and pay thiamine as a toll to the growing hyphae. The simultaneous spatial and metabolic interactions, which are the

bacterial dispersal on fungal highway and sharing of thiamine, establish a mutualism that facilitates the communicating fungal and bacterial species to obtain environmental niche and nutrient, respectively (Fig 5C). Although the bacterial–fungal combination we tested is an artificial condition, the example of co-isolated bacterial–fungal species from nature supports the ecological relevance of the mutualism through fungal highway and share of thiamine.

It has been recently reported that *B. subtilis* supply thiamine to a thiamine-auxotrophic fungus, which is an endophytic fungus colonizing to roots of wide-range plants and has lost genes related to thiamine synthesis (Jiang et al, 2018). The different aspect of our finding is that *A. nidulans* can synthesize thiamine on their own but use thiamine from *B. subtilis*. Thiamine is an essential co-factor for central carbon metabolism in all living organisms and is synthesized by bacteria, fungi, and plants (Jurgenson et al, 2009). Because thiamine often limits the growth of these organisms, they have evolved numerous strategies to obtain thiamine from the natural environment (Kraft & Angert, 2017). Thiamine riboswitches are one of the strategies used to tightly regulate thiamine synthesis and uptake (Cressina et al, 2011). Because we find the bacteria secrete thiamine extracellularly, the neighboring non-auxotrophic bacteria and fungi, such as *A. nidulans*, in nature could use the thiamine by uptake to save the cost rather than by synthesis. Thiamine and riboswitch have the potential to be used widely and universally stimulating symbiosis among microbes and even inter-kingdom interactions in nature. Other vitamins besides thiamine, which are essential for growth but sufficient in small amounts, are likely the seeds of commensal and mutualistic interactions of microbes in nature (Klein et al, 2013; Magnúsdóttir et al, 2015; Palacios et al, 2016; Sokolovskaya et al, 2020).

The affinity of fungal–bacterial interactions is selective depending on the combination of species (our unpublished data). Besides natural auxotrophy, secondary metabolites are also involved in microbial communication for selective interaction. Especially, soil-dwelling bacteria and fungi produce a wide range of secondary metabolites, which function as communication signals among microorganisms to compete and interact with others (Macheleidt et al, 2016). Some secondary metabolite genes are up-regulated in *A. nidulans* co-cultured with *B. subtilis*. The combined analysis of natural auxotrophy and secondary metabolites in co-culture of bacteria and fungi will provide hints to understand selective microbial communication. In addition, live imaging of bacterial–fungal co-culture represents an efficient approach to bioassays that screen for affinities between bacteria and fungi. Recent studies indicate coordinated interactions between fungi and bacteria in various situations, such as promotion of plants growth, fermentation, biomass degradation, plant pathogenesis, and human pathogenesis (Nazir et al, 2010; Frey-Klett et al, 2011; Barkal et al, 2017; Zhang et al, 2018). Although most studies reveal the metabolic interactions, besides them, imaging the localization and functional distribution of microbes is increasing in importance.

# Materials and Methods

## Strains and media

A list of *A. nidulans* and *B. subtilis* strains used in this study is given in Table S4. Supplemented minimal medium for *A. nidulans* and standard strain construction procedures were described previously (Takeshita et al, 2013). If necessary, thiamine was complemented at 10 $\mu$M. Surface soil sample was collected at the depth of (0–5 cm) in secondary forest in University of Tsukuba, Japan. The soil sample was gently sieved by 250-$\mu$m stainless steel mesh in a field moist condition.

## Strain construction

*B. subtilis* Δ*thi* and Δ*hag* strains were constructed as follows. 500 bp of flanking regions were amplified by PCR using primer sets tenA-5/-N3 and tenA-C5/-3, and hag-N5/-N3 and hag-C5/-3, respectively (Table S5). Antibiotic resistance genes $Cat^R$ and $Spc^R$ were amplified using the primer sets cat-Fw/-Rv and spc-Fw/-Rv. The three fragments were ligated by overlap PCR. The DNA fragments were then transformed into *B. subtilis* 168 to construct Δ*thi* [$Cat^R$] and Δ*hag* [$Spc^R$] strains. The deletion of target genes was confirmed by PCR and sequencing. *B. subtilis* expressing green or red fluorescent protein maintains the plasmid, pHY300-Pveg-ZsGreen-term [$Tet^R$] or pHY300-Pveg-mCherry-term [$Tet^R$] (Toyofuku et al, 2017). *B. subtilis* thiamine reporter strain (thi-rep) maintains the plasmid, pHY300MK, expressing ZsGreen under the constitutive Pveg and mScarlet-1 under the promoter of *tenA-operon* 300 bp amplified using the primer set tenA-5/-mSca-Rv.

## Microscopy

A confocal laser scanning microscope LSM880 (Carl Zeiss) equipped with a 63×/0.9 numerical aperture Plan-Apochromat objective and a 40×/0.75 numerical aperture IR Achroplan W water immersion objectives (Carl Zeiss), were used to acquire confocal microscopic images. Bacteria and/or fungi were irradiated with 488- and 633-nm lasers to detect the Green fluorescent protein (ZsGreen) and reflected light, respectively. Acquired confocal images were analyzed using ZEN Software (Version 3.5; Carl Zeiss) and ImageJ software. Another confocal laser scanning microscope TCS SP8 Tandem scanner 8 kHz (Leica) equipped with HC PL APO 20×/0.75 IMM CORR CS2 objective lens was used to acquire high-speed confocal dual-color microscopic images. Bacteria and fungi were irradiated with 488- and 522-nm lasers to detect the GFP and mCherry, respectively. Epi-fluorescent inverted microscopy: cells were observed with an Axio Observer Z1 (Carl Zeiss) microscope equipped with a Plan-Apochromat 63 × 1.4 Oil or 10 or 20 times objective lens, an AxioCam 506 monochrome camera and Colibri.2 LED light (Carl Zeiss). Temperature of the stage was kept at 30°C by a thermo-plate (TOKAI HIT). Using zoom microscopy, plates were observed by AXIO Zoom V16 and HXP 200C illuminator (Carl Zeiss). Images were collected and analyzed using the Zen system (Carl Zeiss) and ImageJ software.

## RNA-seq analysis

Total RNA was isolated from fungal and bacterial cells that were co-cultured for 8 h in six-well plates. For bacterial RNA isolation, the cells were disrupted using glass beads. For fungal RNA isolation, cells were frozen and homogenized using mortar and pestle, and then the total RNA was extracted using an RNA isolation kit (RNeasy Mini Kit; QIAGEN). Ribosomal RNA was depleted using Ribo-ZERO magnetic kit (Epicentre). The transcripts were fragmented and used as templates to generate strand-specific cDNA libraries by TruSeq Stranded Total RNA LT Sample Prep kit (Illumina). Each sample was

sequenced using 100-bp paired-end reads on an Illumina HiSeq 2500 instrument. Macrogen Inc. supported library preparation, sequencing and partial data analysis. The reads were mapped to reference genomes of *B. subtilis* NCIB3610 (CP020102.1) or *A. nidulans* TN02A3 (GCA_000149205.1) with Bowtie 2 aligner. Read count per gene was extracted from known gene annotations with HTSeq program. After $\log_2$ transformation of RPKM+1 and quantile normalization, differentially expressed genes were selected on conditions of $\log_2 > 2$ in expression level.

### Extraction of thiamine

Monocultures of the *A. nidulans* or *B. subtilis* strains and co-cultures were grown in the minimal medium 200 ml with 0.4*g* shaking at 30°C for 3 d prior extraction. Thiamine extraction from the cells: the cells were sieved using Mira cloth and freeze dried. Then they were frozen with liquid nitrogen and crushed. The resultant pellet was dissolved in 5 ml 0.1 M HCl and heated at 100°C for 15 min. It was filtered and allowed to cool. The filtrate was then freeze dried. Finally, freeze-dried pellet was dissolved in 200 $\mu$l of pre-prepared solvent (10 mM ammonium formate + 1% methanol + 0.1 $\mu$l formic acid). Thiamine extraction from the supernatants: the supernatant was collected after centrifuge 5,800*g* for 5 min and freeze dried. The resultant pellet was dissolved in the pre-prepared solvent.

### $^{13}$C labeling

*B. subtilis* wild-type or $\Delta thi$ cells were cultured in the minimal medium containing [U-13C6, 99%] labeled D-glucose (1%) (Cambridge Isotope Laboratories, Inc.) in 200 ml with 0.4*g* shaking at 30°C for 2 d. Simultaneously, *A. nidulans* were cultured in minimal medium containing normal D-glucose in 200 ml for 2 d. The *B. subtilis* cells were collected by centrifuge, whereas the *A. nidulans* cells were collected using Mira cloth and washed. Then they were co-cultured in minimal medium containing D-glucose 200 ml for 2 d. The fungal cells were sieved through Mira cloth and washed thoroughly with milliQ water to remove any bacterial cells attached to the surface. Extraction of labeled thiamine from the cell extract followed the same protocol as of extraction of thiamine described above.

### LC-MS-MRM analysis

Resultant pellets that were dissolved in the pre-prepared solvent were analyzed by LC–MS (LCMS 8030; Shimadzu) equipped with a 250 × 3.0 mm COMOSIL HILIC Packed Column (particle size 5 $\mu$m; Nacalai Tesque, Inc.). The initial mobile phase was with a ratio of solvent A: solvent B (solvent A—acetontrile:10 mM ammonium acetate in water [9:1]; solvent B—100% acetonitrile), increased to 100% and maintained at that ratio for 7 min. UV/Vis spectra were monitored by SPD-M30A (Shimadzu). The mass spectrometer was operated in MRM mode for quantitative analysis of thiamine in the corresponding samples. Mass spectra were acquired with the following conditions: capillary voltage 4.5 kV; detection range m/z 122 for normal thiamine and 128 (precursor m/z 277) for $^{13}$C-labeled thiamine; desolvation line 250°C; heat block 400°C; nebulizer

gas 3 liters/min; drying gas 15 liters/min. Calibration curves were obtained using the LabSolution software (v5.91 Shimadzu Corporation).

### Fungal biomass

Monoculture of *A. nidulans* and co-cultures with *B. subtilis* wild-type and $\Delta thi$ strains were grown in the minimal medium 100 ml with 0.4*g* shaking at 30°C for 3 d. The cultures were then filtered using Mira cloth, and the pellet was washed with milliQ water thoroughly (to remove the bacteria in the co-culture filtrates). The resultant pellets were then freeze dried (SCANVAC COOLSAFE; LaboGene). The weight of the dried pellets was measured several times in between freeze drying until the weight was constant.

### Bacterial genomic DNA extraction

Monoculture and co-cultures were grown on cellophane film on the minimal medium agar (point inoculation of OD$_{600}$ = 0.01). The plates were incubated at 30°C for 3 d. The cellophane films were washed with sterilized milliQ carefully and thoroughly to collect the bacterial cells. The collected samples were subjected to bacterial genomic DNA extraction protocol using Wizard Genomic DNA purification kit. The purified genomic DNA was quantified using NanoDrop (Thermoscientific nanodrop 2000; Thermo Fisher Scientific).

### MATLAB

ImageJ was used to generate a list of cell centroid positions ($x_i$, $y_i$), where $i$ is the frame index number. MATLAB was used to calculate the instantaneous speed of each cell and then speed heat maps were generated. These heat maps were plotted as function of instantaneous position (see Fig 1C), whereas speed was plotted as a function of time. The underlying motion was extracted using a moving window average of five consecutive values to smooth the data and then fitted using a sinusoid.

### Statistical analysis

*t* tests were used to evaluate the mean difference between two sets.

## Data Availability

All data generated or analyzed during this study were included in the manuscript and Supplementary Information. Total RNAseq data collected has been uploaded as the source data accompanying the manuscript. The original data generated during and/or analyzed during this study are available from the corresponding author on reasonable request.

## Supplementary Information

# Acknowledgements

We appreciate the experimental support of Hiroko Kato and Takamitsu Soma at the University of Tsukuba. This work is supported by the Japan Society for the Promotion of Science (JSPS) KAKENHI grants 18K05545 and 18K15143, a grant from the Institute for Fermentation, Osaka (IFO), Noda Institute for Scientific Research Grant and Japan Science and Technology Agency (JST) ERATO grant number JPMJER1502.

## Authors Contributions

G Abeysinghe: formal analysis, validation, investigation, and visualization.
M Kuchira: formal analysis, validation, investigation, and visualization.
G Kudo: formal analysis, validation, investigation, and visualization.
S Masuo: formal analysis, validation, investigation, and methodology.
A Ninomiya: investigation and methodology.
K Takahashi: software and formal analysis.
AS Utada: data curation, software, and formal analysis.
D Hagiwara: data curation and formal analysis.
N Nomura: conceptualization, resources, funding acquisition, and project administration.
N Takaya: conceptualization, resources, project administration, and writing—review and editing.
N Obana: conceptualization, resources, data curation, formal analysis, funding acquisition, validation, investigation, visualization, methodology, project administration, and writing—original draft, review, and editing.
N Takeshita: conceptualization, resources, data curation, formal analysis, supervision, funding acquisition, validation, investigation, visualization, methodology, project administration, and writing—original draft, review, and editing.

## Conflict of Interest Statement

The authors declare that they have no conflict of interest.

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
