## [Reviewer comments · Life Science Alliance]

Fungal mycelia and bacterial thiamine establish a mutualistic growth mechanism

Gayan Abeysinghe, Momoka Kuchira, Gamon Kudo, Shunsuke Masuo, Akihiro Ninomiya, Kohei Takahashi, Andrew Utada, Daisuke Hagiwara, Nobuhiko Nomura, Naoki Takaya, Nozomu Obana, and Norio Takeshita

DOI: <https://doi.org/10.26508/lsa.202000878>

Corresponding author(s): Norio Takeshita, University of Tsukuba and Nozomu Obana, University of Tsukuba

Review Timeline:

Submission Date:	2020-08-16
Editorial Decision:	2020-08-18
Revision Received:	2020-08-20
Editorial Decision:	2020-08-21
Revision Received:	2020-08-25
Editorial Decision:	2020-09-04
Revision Received:	2020-09-07
Accepted:	2020-09-14

Scientific Editor: Shachi Bhatt

Transaction Report:

Please note that the manuscript was previously reviewed at another journal and the reports were taken into account in the decision-making process at Life Science Alliance. Since the original reviews are not subject to Life Science Alliance's transparent review process policy, the reports and author response cannot be published.

August 18, 2020

Re: Life Science Alliance manuscript #LSA-2020-00878-T

Dr. Norio Takeshita
University of Tsukuba
Faculty of Life and Environmental Sciences
Tennodai 1-1-1
Tsukuba 305-8572
Japan

Dear Dr. Takeshita,

Thank you for submitting your manuscript entitled "Fungal mycelia and bacterial thiamine establish a mutualistic growth mechanism" to Life Science Alliance (LSA). The manuscript was assessed by LSA editors and deemed appropriate for publication at LSA with minor revisions.

For a brief overview: The manuscript was peer-reviewed at another journal and the authors chose to transfer the reviewer comments, their point-by-point response, and the revised manuscript to LSA. The main criticisms raised by the reviewers pertained to conceptual advance and a hazy understanding of the relationship between the bacterium and the fungus about thiamine production, uptake and use. Given that this is the first time such a mutualistic relationship has been shown between a non-auxotrophic fungi and *B. subtilis*, the advance was considered to be sufficient for LSA.

We encourage you to re-submit your revised manuscript to LSA with the following minor edits:

- + The abstract needs to be improved for clarity and sentence structure (sentence 1, sentence 3, last line)
- + please elaborate the discussion about the the complex relationship b/w the bacterium and the fungus about thiamine production, uptake and use in the revised manuscript

Thank you for this interesting contribution to Life Science Alliance. We are looking forward to receiving your revised manuscript.

Sincerely,

Shachi Bhatt
Executive Editor
Life Science Alliance

B. MANUSCRIPT ORGANIZATION AND FORMATTING:

August 21, 2020

RE: Life Science Alliance Manuscript #LSA-2020-00878-TR

Dr. Norio Takeshita
University of Tsukuba
Microbiology Research Center for Sustainability (MiCS)
Tennodai 1-1-1
Tsukuba 305-8572
Japan

Dear Dr. Takeshita,

Thank you for submitted your revised manuscript entitled "Fungal mycelia and bacterial thiamine establish a mutualistic growth mechanism". We would be happy to publish your paper in Life Science Alliance (LSA) pending final revisions necessary to meet our formatting guidelines.

Along with the points mentioned below, please also make the following edits so as to meet LSA's formatting guidelines:

Order of manuscript parts -

- References need to be written as 10 authors et al in the Reference list
- add supplemental figure legends right after the figure legends
- add supplemental table legends in the manuscript after the supplemental figure legends
- add movie legends in the manuscript after the supplemental table legends
- add supplemental references to the main list of references

Manuscript edits required:

- Large datasets, sequences should be deposited in one of the relevant public databases and the accession codes should be included in the data availability section
- Is the "unpublished data" mentioned on Pg 10 from the same lab as this paper? If it is from a different lab, please provide an authorization in writing by those involved and submit it to the editorial office.

Callouts -

- Please add callouts for Fig 4C, 4E, S1B, S6C,

Figures -

- Figure 2D the box in the 2D 17h does not perfectly match with the zoomed in image on the right - please place the dotted box correctly
- The spacing between Fig 4A and 4B needs to be improved so that there is a clearer distinction between the 2 panels
- Add scale bar for Fig S4B
- Figure S7A - the box in S7A (B+F Thi-) does not exactly match with the zoomed in images below - please place the dotted box correctly
- Please provide supplemental figures as 1 file & 1 page per figure

Tables -

--Table S2 and S3 need to be in portrait format

Figures & Fig legends match? -

--Some parts of the manuscript say "Fig.", some say "Figure", some say "Fig"- please reconcile

--Fig 1 legend - For Figure 1A - please clarify that the green fluorescence seen corresponds B. subtilis

--Fig 2 legend - for Figure 2B - please specify what the dotted white arrow represents in the legend

Source Data is missing

Guidelines for source data: One file per figure, or per figure panel. If there are several files per figure panel then they must be Zipped together, no more than one file per figure panel. Source data for Supplementary figures need to be placed in a folder per figure and ZIPped all together (using zipr). ZIP files need to be uploaded as file type 'Supplemental Material'.

Additional information on Author guidelines can be found here: <http://www.life-science-alliance.org/authors>

A. FINAL FILES:

B. MANUSCRIPT ORGANIZATION AND FORMATTING:

Full guidelines are available on our Instructions for Authors page, <http://www.life-science->

alliance.org/authors

Sincerely,

Shachi Bhatt
Executive Editor
Life Science Alliance

September 4, 2020

RE: Life Science Alliance Manuscript #LSA-2020-00878-TRR

Dr. Norio Takeshita
University of Tsukuba
Microbiology Research Center for Sustainability (MiCS)
Tennodai 1-1-1
Tsukuba 305-8572
Japan

Dear Dr. Takeshita,

Thank you for submitting your revised manuscript entitled "Fungal mycelia and bacterial thiamine establish a mutualistic growth mechanism". As we were preparing the manuscript for publication, we realized that some formatting issues still remained. Please address the following to comply with Life Science Alliance's guidelines:

- + Please include Tables and their legends right after the supplemental legends in the main manuscript text file (<https://www.life-science-alliance.org/manuscript-prep#format>)
- + Please include a Author contributions section after Acknowledgements
- + LSA requires large datasets, RNA-seq, mass-spec, to be submitted to public databases, please do so and include the accession number (<https://www.life-science-alliance.org/manuscript-prep#datadepot>)
- + Please include a callout for Fig 4E in the manuscript text
- + The dotted box placed in Figure 2D 17H still does not perfectly match with the zoomed image, please correct
- + Please provide a scale bar for Figure S4B
- + Please combine the Supplemental Figure references with the main text references
- + Please clarify why the Figure S1B and Fig S5 were removed from the revised manuscript. Since the reviewers saw the paper with those figures in, we would require you to include those figures in the published version as well.

To upload the final version of your manuscript, please log in to your account:
<https://lsa.msubmit.net/cgi-bin/main.plex>

****The license to publish form must be signed before your manuscript can be sent to production. A**

link to the electronic license to publish form will be sent to the corresponding author only. Please take a moment to check your funder requirements.**

Sincerely,

Shachi Bhatt, Ph.D.
Executive Editor
Life Science Alliance

September 14, 2020

RE: Life Science Alliance Manuscript #LSA-2020-00878-TRRR

Dr. Norio Takeshita
University of Tsukuba
Microbiology Research Center for Sustainability (MiCS)
Tennodai 1-1-1
Tsukuba 305-8572
Japan

Dear Dr. Takeshita,

Thank you for submitting your Research Article entitled "Fungal mycelia and bacterial thiamine establish a mutualistic growth mechanism". It is a pleasure to let you know that your manuscript is now accepted for publication in Life Science Alliance.

DISTRIBUTION OF MATERIALS:

Congratulations on this very interesting paper. I hope you found the editorial process at Life Science Alliance to be constructive. We look forward to future exciting submissions from your lab.

Sincerely,

Shachi Bhatt, Ph.D.
Executive Editor
Life Science Alliance